# Microfiber release from real soiled consumer laundry and the impact of fabric care products and washing conditions

**Neil J. Lant**[1]*, **Adam S. Hayward**[1], **Madusha M. D. Peththawadu**[2], **Kelly J. Sheridan**[2], **John R. Dean**[2]

**1** Procter & Gamble, Newcastle Innovation Center, Newcastle upon Tyne, United Kingdom, **2** Department of Applied Sciences, Northumbria University, Newcastle upon Tyne, United Kingdom

* lant.n@pg.com

**Data Availability Statement:** All relevant data are within the manuscript and its Supporting Information files.

## Abstract

Fiber release during domestic textile washing is a cause of marine microplastic pollution, but better understanding of the magnitude of the issue and role of fabric care products, appliances and washing cycles is needed. Soiled consumer wash loads from U.K. households were found to release a mean of 114 ± 66.8 ppm (mg microfiber per kg fabric) (n = 79) fibers during typical washing conditions and these were mainly composed of natural fibers. Microfiber release decreased with increasing wash load size and hence decreasing water to fabric ratio, with mean microfiber release from wash loads in the mass range 1.0–3.5 kg (n = 57) found to be 132.4 ± 68.6 ppm, significantly (p = 3.3 x $10^{-8}$) higher than the 66.3 ± 27.0 ppm of those in the 3.5–6.0 kg range (n = 22). In further tests with similar soiled consumer wash loads, moving to colder and quicker washing cycles (i.e. 15°C for 30 mins, as opposed to 40°C for 85 mins) significantly reduced microfiber generation by 30% (p = 0.036) and reduced whiteness loss by 42% (p = 0.000) through reduced dye transfer and soil re-deposition, compared to conventional 40°C cycles. In multicycle technical testing, detergent pods were selected for investigation and found to have no impact on microfiber release compared to washing in water alone. Fabric softeners were also found to have no direct impact on microfiber release in testing under both European and North American washing conditions. Extended testing of polyester fleece garments up to a 48-wash cycle history under European conditions found that microfiber release significantly reduced to a consistent low level of 28.7 ± 10.9 ppm from eight through 64 washes. Emerging North American High-Efficiency top-loading washing machines generated significantly lower microfiber release than traditional top-loading machines, likely due to their lower water fill volumes and hence lower water to fabric ratio, with a 69.7% reduction observed for polyester fleece (n = 32, p = 7.9 x $10^{-6}$) and 37.4% reduction for polyester T-shirt (n = 32, p = 0.0032). These results conclude that consumers can directly reduce the levels of microfibers generated per wash during domestic textile washing by using colder and quicker wash cycles, washing complete (but not overfilled) loads, and (in North America) converting to High-Efficiency washing machines. Moving to colder and quicker cycles will also indirectly reduce microfiber release by extending the lifetime of clothing, leading to fewer new garments being purchased and

**Funding:** This work was sponsored by Procter & Gamble, a manufacturer of detergents and fabric softeners. The funder provided support in the form of salaries for N.J.L. and A.S.H. and decision to publish but did not have any additional role in the study design, data collection and analysis or preparation of the manuscript. The specific roles of these authors are articulated in the 'author contributions' section.

**Competing interests:** NJL and ASH are employed by Procter & Gamble. This does not alter our adherence to PLOS ONE policies on sharing data and materials.

hence lower incidence of the high microfiber release occurring during the first few washes of a new item.

## Introduction

### Role of laundry in the microfiber issue

Microplastic pollution, and its impact on marine ecosystems, is now recognized as a significant environmental issue [1]. The formulation of plastic microbeads in consumer products was initially considered to be an important cause, although more recent findings suggest that microfibers generated during washing of synthetic textiles and released to waste water is likely to be a more significant contributor [2–4]. Analysis of fiber contamination in European marine sediments suggests that the issue is not restricted to synthetic fibers with detection of significant quantities of cellulosic fibers [5], despite natural fibers such as cotton and rayon being significantly more biodegradable than synthetic fibers such as polyester [6]. Recent analysis of fibers released during tumble drying [7] suggests that the environmental impact of contaminants present in or on the microfibers might also be an important consideration.

### Potential solutions

Several textile design principles have been established which reduce microfiber release during laundering such as avoiding loose knits and fleeces [8], and more research is being undertaken to complete understanding and encourage their adoption by the textile industry. Many textile construction parameters are relevant [9] including polymer type, polymer origin, yarn size, yarn length, brightness, twist, fabric construction, fabric finishing, and processes used for cutting, sewing, storage, washing and drying. Application of finishing treatments such as the ElectroFluidoDynamic (EFD) method [10] or pectin-based finishes [11] were also found to significantly reduce microfiber release.

Installation of washing machine filters is also proposed to mitigate the issue [5], with the Lint LUV-R system showing high efficiency, especially when compared to the 'Cora ball' device [12]. However, although appliance manufacturers such as Arçelik A.Ş. (Istanbul, Turkey) are beginning to introduce washing machine filters [13], it is not yet clear whether the whole industry will follow on a global basis, and this is unlikely to be a complete solution given that much of the world's population does not use a washing machine and filtration is not 100% efficient [13].

Removal of microfibers at wastewater treatment plants is also being considered, although this also challenging as the microfibers would need to be eliminated from both effluent and sludge to prevent release into the environment [14,15].

### Impact of fabric care products and washing conditions

Changing consumers' behavior relating to the purchase and care of clothing has the potential to rapidly reduce textile microfiber pollution. Therefore, several groups have been studying the impact of home laundering on microfiber release phenomena, as described in recent reviews [16,17]. While there is consensus in some areas, for example that washing in colder water or quicker cycles reduces microfiber release, apparently conflicting conclusions have been reported in other areas such as the impact of detergent. Hernandez et al. [18] found that liquid and powder detergents cause increased microfiber release compared to washing in

deionized water. Carney Almroth et al. [8] also reported that detergents cause significantly higher microfiber generation compared to washing in water alone for three out of four fabric samples tested. Napper and Thompson [19] also reported that 'bio-detergent' (enzyme-containing detergents) increased fiber loss in some cases but appeared to decrease or have no impact on fiber loss in others whereas Pirc et al. [20] concluded that the detergent has little impact. Yang et al. [21] found that use of a detergent significantly increased microfiber loss, especially for polyester fibers at lower wash temperatures. Recently, Cesa et al. [22] reported that detergent use can reduce microfiber release from synthetic clothing. Kelly et al. [23] reported that water to fabric ratio was an important factor in microfiber shedding levels, confirmed in both full-scale and model systems.

Napper and Thompson [19] suggested that use of fabric softeners might exacerbate microfiber release under certain conditions, yet De Falco et al. [24] concluded that these products reduce microfiber release by more than 35% whereas Pirc et al. [20] concluded that fabric softeners have no impact. Several studies [19,20,22,25,26] found that new fabrics generate most microfibers, although others [18] reported similar release levels over multiple cycles and one [27] concluded that artificially aged garments exhibit higher microfiber release than their new equivalents.

These apparently conflicting conclusions may be caused by differences in test methodology, especially regarding the application of small-scale devices such as the Launder-Ometer® (SDL Atlas), or GyroWash (James Heal Ltd.) or Washtec P (Roaches) which are commonly used for small-scale simulation of washing for some aspects of textile testing such as dye fading or dye transfer. Although De Falco et al. [28] concluded that a GyroWash protocol provided a good estimation of full-scale testing, they did not study the impact of different parameters to fully validate such protocols. Recently, Kelly et al. [23] recommended use of a Tergotometer (Copley Scientific) for small-scale simulated laundry testing, providing evidence of correlation to full scale washing. All previous testing was conducted with clean wash loads, often involving new or artificially aged garments and homogenous loads comprising one or a few types of garment. Studies involving the reality of domestic laundry, with authentic soiled wash loads containing a mixture of garment types, soils, fibers, ages and textures have apparently not yet been published.

In this study we measure the release of microfibers from typical soiled wash loads and the impact of moving to colder, quicker, wash cycles on microfiber generation and whiteness maintenance. We also expand the body of evidence regarding impact of detergent, rinse-added fabric softener, and garment age on microfiber generation using full scale testing involving sets of domestic washing machines typical of those used in Europe and North America. The findings will be relevant to the education of consumers in ways to reduce their environmental impact, to those involved in microfiber research within the textile and appliance industries, and to companies involved in the development and manufacture of fabric care products.

## Materials and methods

### Textiles

European soiled consumer wash loads were sourced from households in Newcastle upon Tyne, U.K. in the summer of 2017 for two separate experiments. Consent forms outlining the specific details of the research and handling of data were signed by all volunteers. These contained a mixture of washable apparel and household textiles, pre-sorted by the consumers into loads as they would for laundering at home. Each load was weighed prior to washing to enable calculation of microfiber release as a proportion of load mass. The washed loads were returned to the consumers immediately after testing. Photographs of typical baskets of soiled laundry are given in S1 Fig.

Clean whiteness monitors (labelled W1-W18) were composed of small (5 cm × 5 cm) swatches of 18 different white T-shirt fabrics listed in S1 Table. These were tagged (using Tach-It® gun, Tach-It, S. Hackensack, N.J., U.S.A.) onto a 27 cm × 30 cm knitted cotton backing cloth supplied by Warwick Equest Ltd. (Consett, U.K.). These swatches were used to measure soil redeposition and dye transfer in testing the impact of moving to colder and quicker washing cycles on microfiber release.

Evaluation of the impact of detergent and fabric softener was conducted using three (European tests) or four (U.S.A. top load tests) red Fruit of the Loom® full zip fleeces (product code 62–510, size XL, 100% pill-resistant polyester, density 300g/m²) per load or 10 black Fruit of the Loom® performance T-shirts (product code 61–390, size XL, 100% textured polyester, density 140g/m²). All new garments, including those used to prepare whiteness monitors listed in S1 Table, were purchased from BTC Activewear Ltd., Wednesbury, U.K.

## Fabric care products

European tests with liquid detergent were conducted using 70 ml Ariel® liquid per wash. European tests with detergent pods were conducted using one Ariel® 3in1 pod per wash.

North American tests were conducted using one Tide® 3in1 detergent pod per wash. European fabric softener tests were conducted with 25 ml Lenor® Spring Awakening. North American fabric softener tests were conducted using 50 ml Ultra Downy® April Fresh. European and North American products were manufactured by Procter & Gamble and sourced from retailers in the U.K., and U.S.A., respectively, during summer 2017.

## Washing procedure

**European testing.**    All European washing tests were conducted using 10 grains per U.S. gallon hardness water and Miele® W3622 washing machines using either the 40˚C Cotton Short (85 minutes total duration, 1600 rpm spin speed) or Cold Express (a quick cycle using unheated water at 15˚C, 30 minutes total duration, 1600 rpm spin speed) programs. All European washing machine tests were conducted using sets of eight identical machines fed by the same water supply. Treatments were rotated between machines and run in quadruplicate, except for the extended wash (48 cycles) test which was run in quadruplicate for the 1, 4 and 8 cycle data and in duplicate for the 16, 32 and 64 cycle data. Fabrics were not dried between wash cycles in testing involving more than one wash cycle.

**North American testing.**    All North American washing tests were conducted using 6 grains per U.S. gallon water and 27˚C water temperature. The North American traditional top-loader washing tests used Kenmore® 600 washing machines with 64 L fill volume and 18-minute Super wash cycle, with one rinse. North American High-Efficiency top-loader washing tests used Whirlpool® Cabrio washing machines with 30 L fill volume and an 18-minute wash (47-minute total cycle) Casual cycle. All North American washing machine tests were conducted using sets of four identical machines fed by the same water supply. Treatments were rotated between machines and run in quadruplicate. Fabrics were tumble dried between cycles using Miele® T8322 tumble dryers at a low temperature setting (50˚C) for 1 h, as tumble drying is very common in the North American market although fibers released at this stage are collected by the dryer and disposed in household trash and hence are not released in waste water.

## Microfiber and fabric analysis

**Collection and mass measurement.**    Microfibers were collected and quantified using the protocol described by Kelly et al. [23] which is based on the method of Napper and Thompson

[19]. Washing machines were thoroughly cleaned prior to testing and confirmed to release no fibers by running a cycle without any garments inside. Water was collected into 25 L high density polyethylene containers from the drain hose of the washing machine for the complete wash and rinsing process. After each wash, any fibers remaining in the machine were washed out and collected by running an additional 'washout' cycle with no detergent after the fabrics were removed. All water from the test run and 'washout' cycle was then filtered through 20 μm CellMicroSieve® (BioDesign Inc., Carmel, N.Y., U.S.A.). Containers were thoroughly rinsed to collect all fibers. The residual fibers collected were washed with clean water, re-suspended in clean water and then filtered onto pre-weighed Whatman® No 541 filter paper (G.E. Life Sciences, Little Chalfont, U.K.) using a Büchner funnel under vacuum before drying overnight at 50˚C. The mass of collected fibers was then calculated, corrected for the percentage loss in filter paper weight on drying which was determined by recording the mean percentage mass loss on drying of 10 similar papers. Microfiber release data is presented in parts per million (ppm), i.e. as mg of released fiber per kg initial dry fabric load mass. Washing machines were thoroughly cleaned between washes using an additional cleaning cycle; the efficiency of this process to prevent cross-contamination of fibers from one run to another was confirmed using the same methodology described in the Quality Assurance section of Kelly et al. [23] which involved the same washing machines as the present study.

**Microfiber analysis.** Fibers were recovered from the center of the filter paper, using a mask of diameter 2 cm (Fig 1). Recovery was achieved by pressing the high-adhesive J-Lar® tape (WA Products, Essex, U.K.) on the 2 cm diameter sampling point. Initially, a single thumb press on the external surface of the J-Lar® tape allowed a localized pressure to be exerted; this was followed by additional pressure by dragging over the same sampling point a 2 kg weight. The dragging of the 2 kg weight allowed a constant pressure to be applied to aid fiber recovery on the J-Lar® tape. The J-Lar® tape was then carefully lifted and attached to a transparent A5 acetate sheet, which had been previously appropriately labelled. The entire recovery process with J-Lar® tape was repeated between 3 and 5 times (Fig 1 shows 4 J-Lar® tape lifts), based on the exhaustive recovery of fibers, as evidenced by negligible fiber recovery on any subsequent filter paper transfer. Then, a fiber lifted J-Lar® tape containing a median-loaded amount of fibers was selected for further analysis and identification; the selection of the median-containing J-Lar® tape was based on visual inspection, by eye, to adequately reflect the distribution of fibers collected on the J-Lar® tape from the 1 to 3/5 micro-sampling points. All J-Lar® lift tape samples were mounted on their own unique labelled transparent A5 acetate sheet.

The selected median J-Lar® fiber tape was subsequently micro-sampled. This micro-sampling of the selected J-Lar® tape sample involved inversion of the tape to allow recovery of fibers with stainless steel HP Grade tweezers, type 5 (TAAB, Reading, U.K.). Micro-sampling was controlled via a selected window of initial diameter 0.1 mm that allowed recovery of 100 fibers; subsequently if 100 fibers were not recovered then the diameter of the sampling window was increased up to 1.0 mm. Recovery of fibers was carried out manually using stainless steel tweezers under a Low Power Microscope (x25 magnification, Leica M60, Leica, Milton Keynes, U.K.). An individual fiber was recovered and placed on a microscope slide containing a drop of Phytohistol® (Fisher Scientific, Loughborough, U.K.) and covered with a glass coverslip (diameter 9 mm, thickness number 1; VWR, Lutterworth, Leicestershire, U.K.); in total, 10 fibers in their inherent Phytohistol® drops were mounted on one microscope slide. In addition, it is noted that all 100 fibers, from the initial sample point, were randomly collected irrespective of color, size and shape i.e. no identification had been done at this stage, but excluded colorless fibers. Then initial fiber identification was carried out using a combination of High Power Microscope (x 400 magnification, Olympus CX23, Olympus, Hamburg, Germany) and

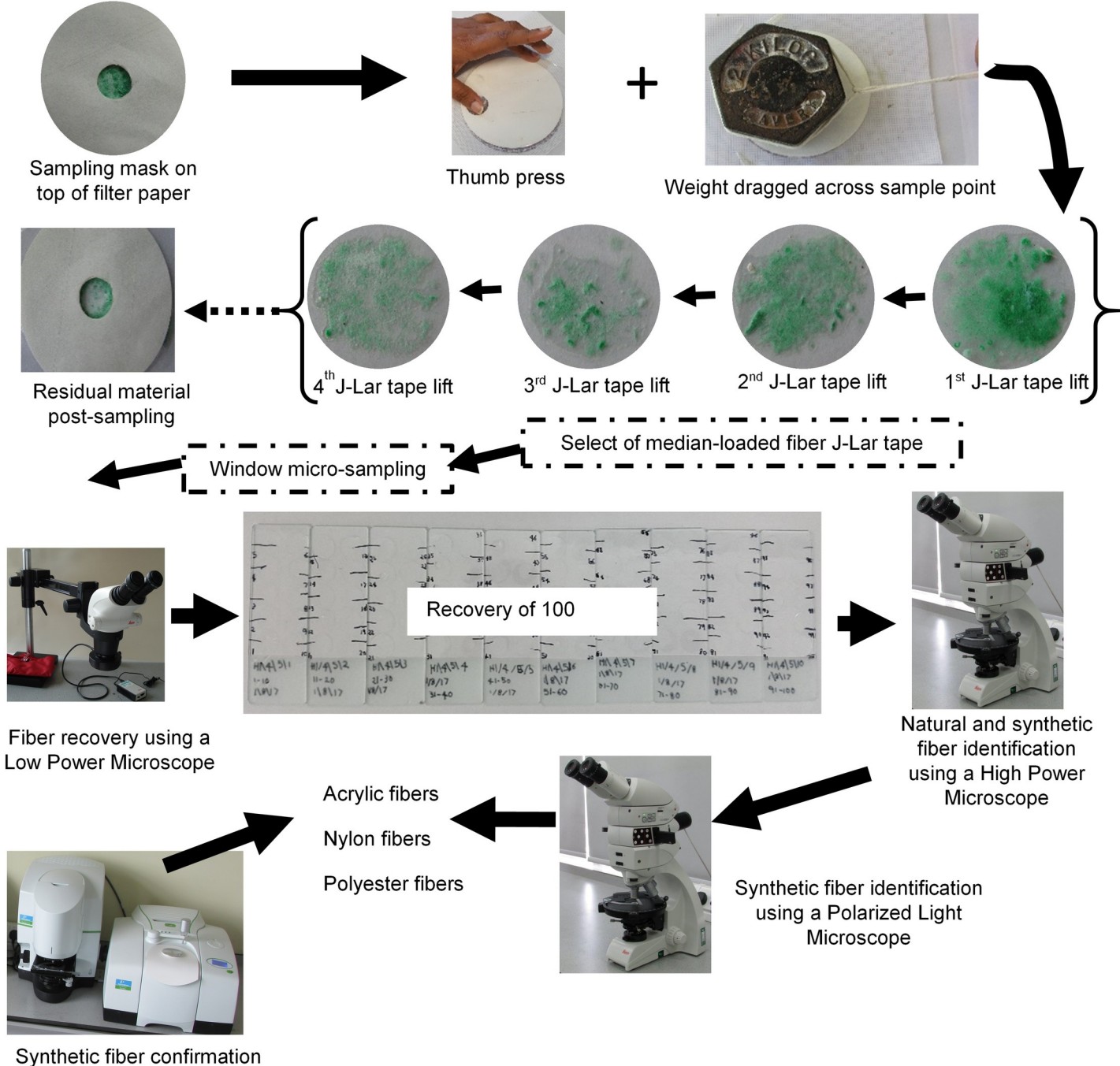

**Fig 1. Systematic process for colored fiber recovery and identification.**

Polarizing Light Microscopy (Ortholux, Leica, Milton Keynes, U.K.). Using visual identification of surface morphology of shape and characteristic features allowed the discrimination of fiber types e.g. 'synthetic fibers' (i.e. delusterant particles) from those of cotton (i.e. ribbon) and wool (i.e. scales). Further analysis of the 'synthetic fibers', using a Polarized Light Microscope (allowed generic identification of acrylic, nylon or polyester fibers. This was achieved by determining the birefringence value [29] that correlates to fiber type.

Subsequently fiber identification was confirmed using an FT-IR Microscope (Spotlight 150i) coupled to a FT-NIR Spectrometer (Frontier) (Perkin Elmer, Beaconsfield, UK) which had previously been used to characterize fiber type using 100% authentic samples (i.e. acrylic, cotton, elastane, nylon, polyester, polyethylene, polypropylene, silk, viscose and wool) which themselves had previously been tape lifted to eliminate fiber cross-contamination.

A total of 25 filter papers were randomly selected from 79 consumer washing loads to determine the mass of synthetic fibers retained. The initial acid digestion procedure using sulfuric acid of the filter papers, containing the washing effluent, dissolved the filter papers (cellulose), cotton and viscose fibers while the second digestion with sodium hypochlorite dissolved wool [30]. Preliminary testing of the method [30] was done using a range of 100% cotton, viscose, wool, acrylic, nylon and polyester materials; the results validated the identified method on specific fabric samples (as well as digestion of the filter paper). The procedure was then applied to the sampled filter papers from the real consumer fabric test loads.

Filter paper containing the washing machine effluent was cut, using scissors, in to eight pieces and placed in a 250 ml beaker to which 150 ml of 75% $H_2SO_4$ (Analytical grade; Fisher Scientific, Loughborough, U.K.) was added. The beaker was left to stand in a water bath at $30°C \pm 2°C$ for $45 \pm 5$ mins and stirred with a glass rod every 5 mins. The resultant solution, and the residual solids, were quantitatively transferred using additional distilled water for a period of 10 mins, through a 180 μm stainless steel sieve. The solid residue was quantitatively transferred using tweezers, to a 250 ml beaker and 150 ml of 5% sodium hypochlorite (Fisher Scientific, Loughborough, U.K.) added to digest the wool. Again, the beaker was stood in the water bath at $30°C \pm 2°C$ for $45 \pm 5$ mins and stirred every 5 mins. The final residue was collected on a 180 μm stainless steel sieve, with additional rinsing with distilled water for 10 mins. This final residue was collected by quantitatively transferring in to a 100 ml beaker with water and the wash filtrate was vacuum filtered on a pre-accurately weighed Fisher brand 11546873 (diameter 55 mm) filter paper (Fisher Scientific, Loughborough, U.K.) and dried at 110°C for 1.5 h. After drying, the percentage of collected synthetic fibers was then determined, based on the mass of collected fibers as described previously.

**Whiteness monitor analysis.**   After the wash, whiteness monitor fabrics W1-W18 were left to air dry for 24 h and then measured according to CIEDE2000 using a Gretag Macbeth Color-Eye[®] 7000A (X-Rite Ltd, Manchester, U.K.) spectrophotometer to determine the color difference between unwashed and washed fabric. $\Delta E_{2000}$ values were calculated using Eq 1 [31] for each whiteness monitor using the differences in lightness (L´), chroma (C´) and hue (H´) values between unwashed and washed fabric where $k_L S_L$ is a lightness weighting function, $k_C S_C$ is a chroma weighting function, $k_H S_H$ is a hue weighting function, and $R_T$ is a rescaling factor. A whiteness loss of two $\Delta E_{2000}$ units is very noticeable to the eye.

$$\Delta E_{2000} \sqrt{\left(\frac{\Delta L'}{k_L S_L}\right)^2 + \left(\frac{\Delta C'}{k_C S_C}\right)^2 + \left(\frac{\Delta H'}{k_H S_H}\right)^2 + R_T \left(\frac{\Delta C'}{k_C S_C}\right)\left(\frac{\Delta H'}{k_H S_H}\right)} \quad (1)$$

# Results and discussion

## Microfiber release from real European soiled consumer wash loads

**Gravimetric analysis.**   U.K. soiled consumer loads (n = 72) were washed using the same detergent (70 ml Ariel liquid) and European conditions (40°C Cotton Short). See materials and methods section for more details of the general European washing procedure. The mean wash load mass was 3.13 ± 1.02 kg. A mean of 114 ± 66.8 ppm (range 18.6–368.3 ppm) microfiber release was measured and light microscopy images of typical filter papers are given in S2 Fig. Load masses and microfiber release data for each of the wash loads is given in S2 Table

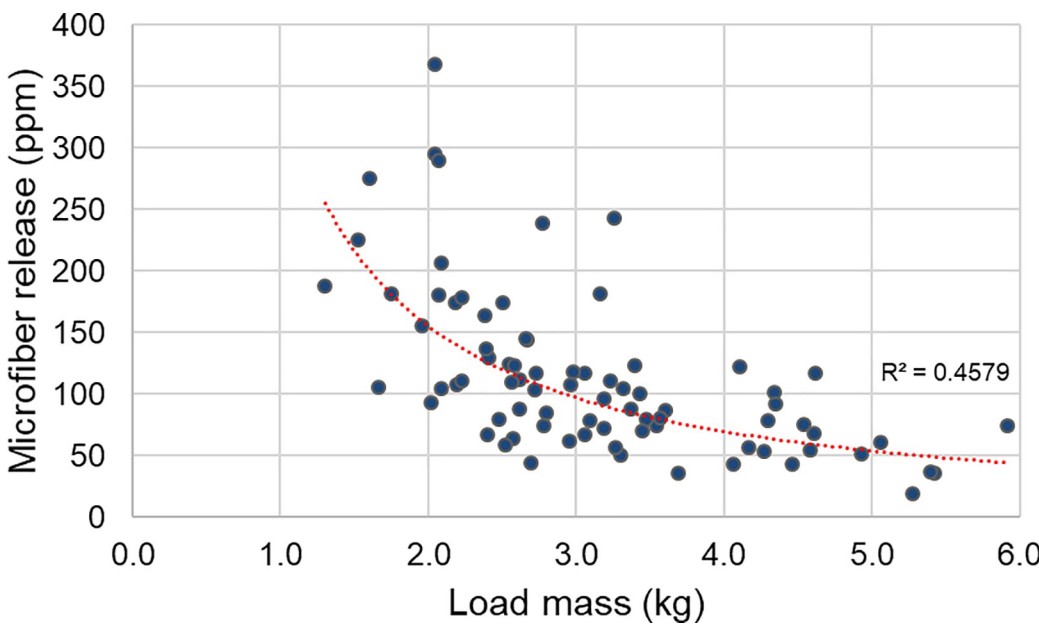

**Fig 2. Correlation of microfiber release from 79 European soiled consumer wash loads against load mass (n = 79).** The trend line relates to the equation $y = 346.23x^{-1.158}$; $R^2 = 0.46$; $p = 7.6 \times 10^{-12}$.

and a standard boxplot of the results is given in S3 Fig. A boxplot summarizes the mean (cross), median (horizontal line within the box), first quartile (low end of the box), third quartile (upper end of the box), upper limit excluding outliers (horizontal line at end of upper whisker), lower limit excluding outliers (horizontal line at end of lower whisker) and outliers (small circles above or below the whiskers). These results are similar to the range of 124–308 ppm observed by De Falco et al. [26] although their tests were mainly focused on new synthetic garments.

According to the International Association for Soaps, Detergents and Maintenance Products (A.I.S.E.) [32], 35.6 billion wash loads are completed across 23 European countries each year, i.e. 1130 washes are started every second. Assuming the load size and release levels in our tests are typical, microfiber release will be around 357 mg/wash (114 ppm x 3.13 kg load) resulting in a total quantity of microfiber release from those 23 countries of around 12,709 tonnes per year.

Fig 2 shows a correlation of wash load mass with ppm microfiber release, with a clear trend towards increasing fiber loss in smaller loads. The trend line relates to the equation $y = 346.23x^{-1.158}$ which has an $R^2$ value of 0.46 and $p = 7.6 \times 10^{-12}$. Of the 79 wash loads tested, 57 were in the mass range 1.0–3.5 kg and these had a mean of 132.4 ± 68.6 ppm microfiber release, compared to the 22 loads in the 3.5–6.0 kg range which had a mean 66.3 ± 27.0 ppm microfiber release, i.e. 50% lower, statistically significant at 99% confidence level (Student's t-test $p = 3.3 \times 10^{-8}$). The smaller the wash load, the higher the water to fabric ratio, which will lead to increased flow of wash solution through the yarns. This suggests that consumers should aim to wash complete loads but use an appropriate quantity of detergent and refrain from overfilling the washing machine as that would have a negative impact on cleaning performance and potentially lead to mechanical failure of the appliance.

**Fiber characterization – acid digestion.** The results (summarized in Fig 3 from data in S3 Table) indicated that a significant portion, i.e. mean 96% (range 90–99%) of the fibers were natural, including cotton, wool and viscose while synthetic fibers, including acrylic, nylon and polyester, accounted for 4% (range 1–10%). This suggests that microfiber release from real

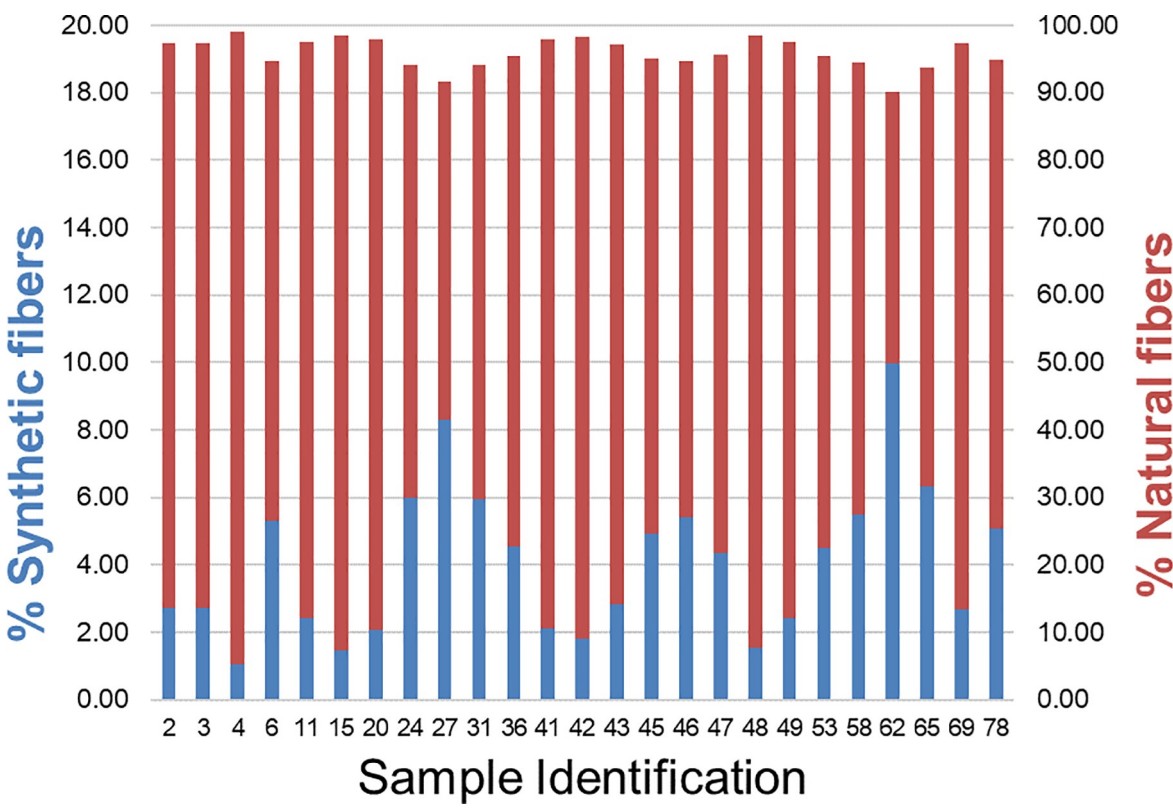

**Fig 3. Percentage composition after fiber digestion.**

consumer loads is dominated by natural fibers, rather than synthetic fibers during the washing process. This can be supported from environmental data [5,33].

**Fiber characterization using optical and FT-IR microscopy.** For the further analysis of fiber classification, three filter papers were randomly selected from 79 consumer washing loads and subjected to a systematic process of colored fiber recovery and identification (Fig 1). Colored fibers were identified as being cotton, wool, viscose or synthetic. However, this subjective procedure only identified the colored fibers and it was noted that several colorless fibers were left unidentified. While the colorless fibers were not identified it can be postulated that they were either cotton (natural) or undyed synthetic fibers. Fibers identified as synthetic were further examined using Polarized Light Microscope (PLM) to determine the generic fiber type [34]. It was found that the colored fiber percentage composition was on average 75%: 25% for natural: synthetic fibers (Table 1). All fiber types identified with optical microscopy were subsequently confirmed using FT-IR microscopy.

**Table 1. Summary of colored fiber identification using optical and FT-IR microscopy.**

| Sample identification | % Natural | | | | % Synthetic | | | |
|---|---|---|---|---|---|---|---|---|
| | Total | Sub-set | | | Total | Sub-set | | |
| | | cotton | viscose | wool | | acrylic | nylon | polyester |
| #1 | 76 | 69 | 2 | 5 | 24 | 7 | 8 | 9 |
| #28 | 71 | 64 | 1 | 6 | 29 | 14 | 6 | 9 |
| #77 | 79 | 61 | 4 | 14 | 21 | 6 | 4 | 11 |
| Average | 75 | | | | 25 | | | |

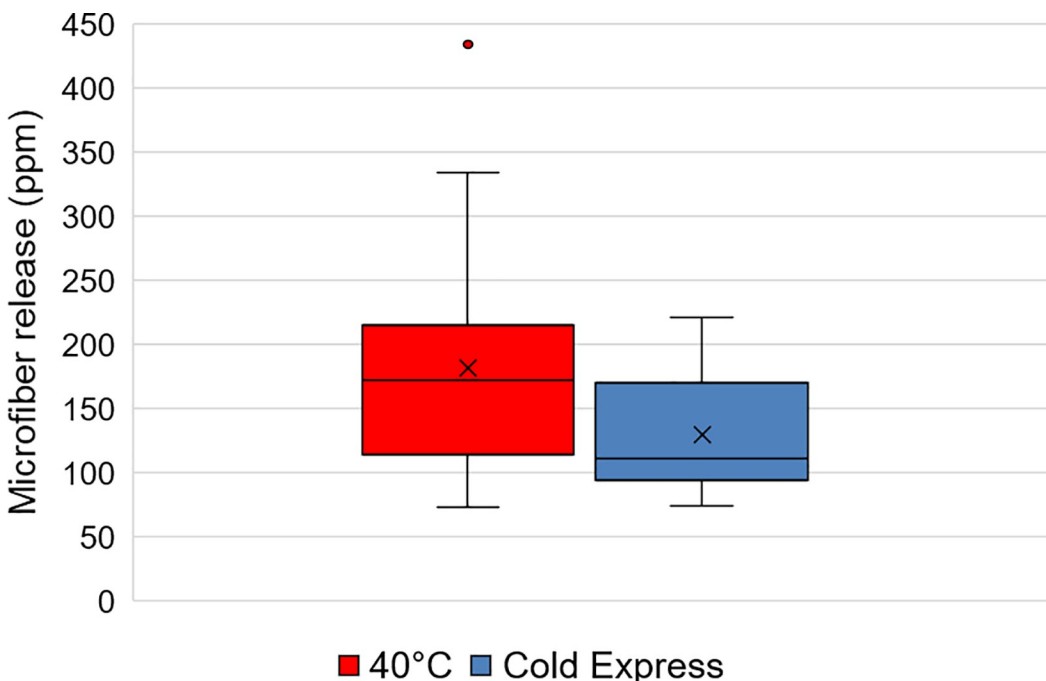

**Fig 4. Boxplot of microfiber release from soiled consumer wash loads in a 40˚C cycle (n = 19) with cold express cycle (n = 19).**

### Impact of moving to cold/quick conditions on microfiber release from real European soiled consumer wash loads

Further work was conducted to measure the impact of consumers moving from a conventional 40˚C wash cycle to emerging cold and quick cycles found on European washing machines. Whiteness monitors were included to measure any impact of colder and quicker washing conditions on dye transfer and soil re-deposition. European soiled consumer loads (n = 38) were washed using the same detergent (70 ml Ariel liquid), with half (n = 19) using typical European conditions (40˚C Cotton Short cycle) and the other half (n = 19) using a colder and quicker cycle (Cold Express). See materials and methods section for more details of the general European washing procedure. The average wash load mass was 2.69 kg for both treatments (± 0.77 kg and ± 0.76 kg for 40˚C and Cold Express cycles, respectively). Fig 4 is a standard boxplot of microfiber release using data from S4 Table, showing a 30% reduction from mean 181.6 ± 87.1 ppm for the 40˚C cycle to 129.5 ± 42.9 ppm for the Cold Express cycle, statistically significant at 95% confidence level (Student's t-test p = 0.036).

The results suggest that temperature and wash duration, two factors known to impact soil removal, are also critical factors in liberation of microfibers from textiles during washing although further work is needed to determine their relative importance.

Whiteness monitors were included in this test to determine the impact of changing washing condition on re-deposition of dyes and soils. These phenomena are important causes of garment integrity loss, leading to the graying of white items and white regions of mixed color items, as well as negatively impacting the shade of colored articles. Fig 5, based on data from S5 Table, compares the average color change ($\Delta E_{2000}$) on each of the 18 whiteness monitors (W1 to W18) washed for a single cycle in the 40˚C and Cold Express conditions. The results show that there is an average 42.3% reduction in whiteness loss on moving from 40˚C to Cold Express cycle, statistically significant at 99% confidence level (Student's t-test p = 0.000). It is

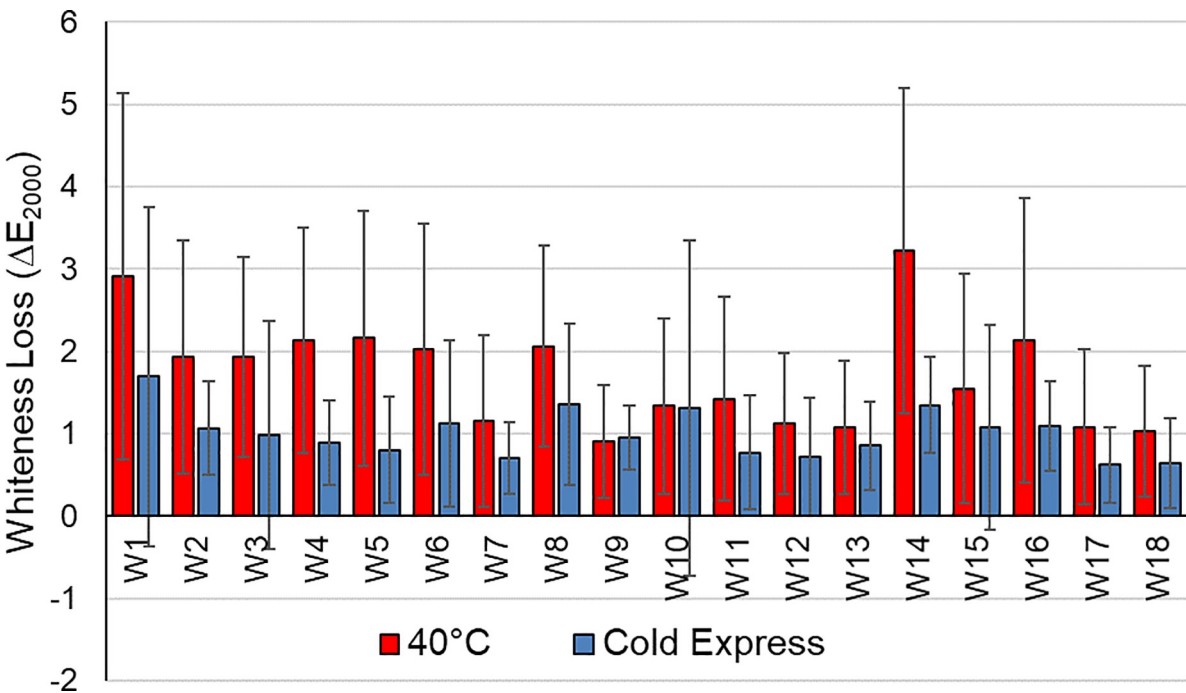

**Fig 5. Comparison of whiteness loss from 40˚C (n = 19) with cold express cycle (n = 19).** Error bars represent the standard deviation.

likely that these improvements in whiteness maintenance arise from multiple mechanisms, particularly reduced dye and soil removal from the fabrics, and reduced redeposition of dyes and soils from the wash solution.

These results confirm that a shift in European consumer behavior away from washing in standard 40˚C cycles towards colder and quicker conditions could significantly reduce microfiber release and slow down other aspects of clothing integrity loss.

## Impact of laundry detergent on microfiber loss

The impact of laundry detergent on microfiber loss was evaluated using full zip fleece garments, over the course of 8 washes with microfiber release measured during the first, fourth and eighth cycle. Washing in the absence of detergent was compared with use of a detergent pod, with testing conducted under European conditions (40˚C Cotton Short cycle). See materials and methods section for more details of the general European washing procedure.

Results summarized in Fig 6 from data in S6 Table show that the use of detergent has no significant impact on microfiber release (Student's t-test p = 0.65), consistent with results published from Pirc et al. [20] but contradicting results published by Hernandez et al. [18] and Carney Almroth et al. [8] which concluded that detergent exacerbated fiber loss. The results also show a significant reduction in microfiber release after the first few cycles, in line with the results from Pirc et al. [20].

The apparent discrepancy between our findings and those of Hernandez et al. [18] and Carney Almroth [8] might be driven by their use of the small-scale simulation devices Washtec P (Roaches) and GyroWash (James Heal Ltd.), respectively. This is corroborated by the good agreement between our findings and those of Pirc et al. [20] which also involved testing in full-scale domestic washing machines. It is likely that these small-scale devices do not adequately replicate the physical processes involved in domestic washing, i.e. they are able to simulate

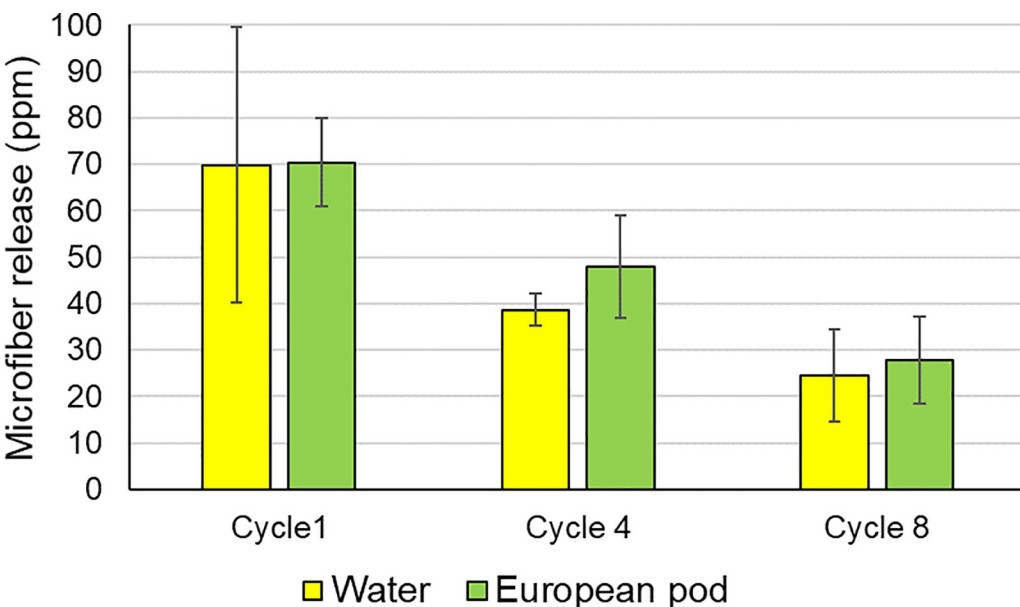

**Fig 6. Impact of using European detergent pod on microfiber release from polyester fleece (n = 24).** Error bars represent the standard deviation.

mainly chemical processes such as action of bleaching ingredients but not more physical processes such as unplucking and breakage of fibers, or release of free microfiber debris from the yarn interiors.

## Impact of fabric softener on microfiber loss and long-term effects

**European testing.** Further European testing using the same full zip fleece garments was conducted to understand the impact of fabric softener use and determine whether the trend of reducing microfiber release with increasing number of wash cycle continues in the longer term. Testing was conducted under European conditions (40˚C Cotton Short cycle). See materials and methods section for more details of the general European washing procedure. The results summarized in Fig 7 from data in S7 Table show that use of fabric softener has no significant impact on microfiber release (Student's t-test p = 0.50). Microfiber release reaches steady low level after eight cycles and does not increase at a later stage in the 48-wash duration of the test. Mean microfiber release for all data between eight and 48 cycles, inclusive, was found to be of 28.7 ± 10.9 ppm. While being consistent with results from Pirc et al. [20], this finding that older garments show less microfiber loss than new garments contradicts previous studies by Hartline et al. [27] although their testing involved an artificial textile aging process involving 24 hours continuous agitation in a traditional North American top-loading washing machine. It is possible that the latter method damages the garments to the extent that fraying occurs, which would be expected to result in increased fiber loss. We did not observe any evidence of such fraying in our 48-cycle study with full-zip fleeces, but it is important to recognize that our tests did not involve any wearing of the garments and this aspect of aging could influence microfiber release.

**North American testing.** Further testing was conducted to show that these conclusions are equally valid in two types of North American top-loading machines using two different garment types. The North American washing machine market has historically been dominated by large top-loading washing machines with a vertical axis screw-like rotating agitator and high wash water volumes (~64 L). These traditional appliances are now being gradually

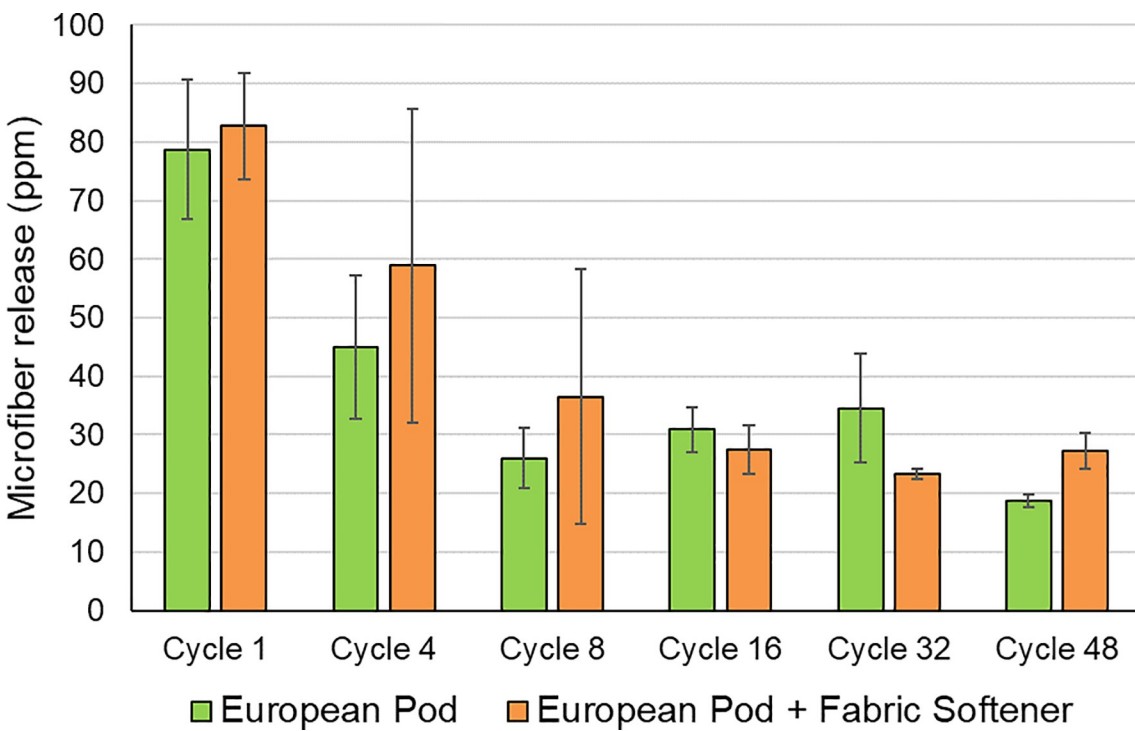

**Fig 7. Impact of fabric softener on microfiber release from polyester fleece: European conditions (n = 36).** Error bars represent the standard deviation.

replaced by High-Efficiency appliances with significantly lower water consumption, with both top-loading and front-loading versions available. Top-loading High-Efficiency appliances use around half the wash water volume of traditional appliances and use a rotating impeller on the base of the drum for agitation. Front-loading High-Efficiency machines have a similar design to European appliances. See materials and methods section for more details of the general North American washing procedure. The results from testing the same full zip fleece and a performance T-shirt in both a traditional U.S.A. washing machine and a High-Efficiency top-loading washing machine are summarized in Fig 8 from data in S8 Table. In line with the European results, fabric softener was not found to have any impact on microfiber release. However, the High-Efficiency top-loading machines caused significantly lower levels of microfiber release compared to the traditional appliance. On average, the full zip fleece and performance T-shirt showed 69.7% (statistically significant; Student's t-test $p = 7.9 \times 10^{-6}$) and 37.4% (statistically significant; Student's t-test $p = 0.0032$) lower microfiber release, respectively, for the High-Efficiency washing machine compared to the traditional appliance. Although High-Efficiency washing machines have several differences compared to traditional machines, including the agitation system, their significantly lower wash water volume and consequently much lower water to fabric ratio is likely to be a key factor in the observed reduction in microfiber release, based on recent testing under European conditions [23]. This difference in water to fabric ratio could be having an impact on the mechanisms of microfiber loss.

## Conclusions

Real consumer wash loads release significant levels of microfibers into water treatment systems, although these are predominantly natural fibers which, being staple fibers, are likely to

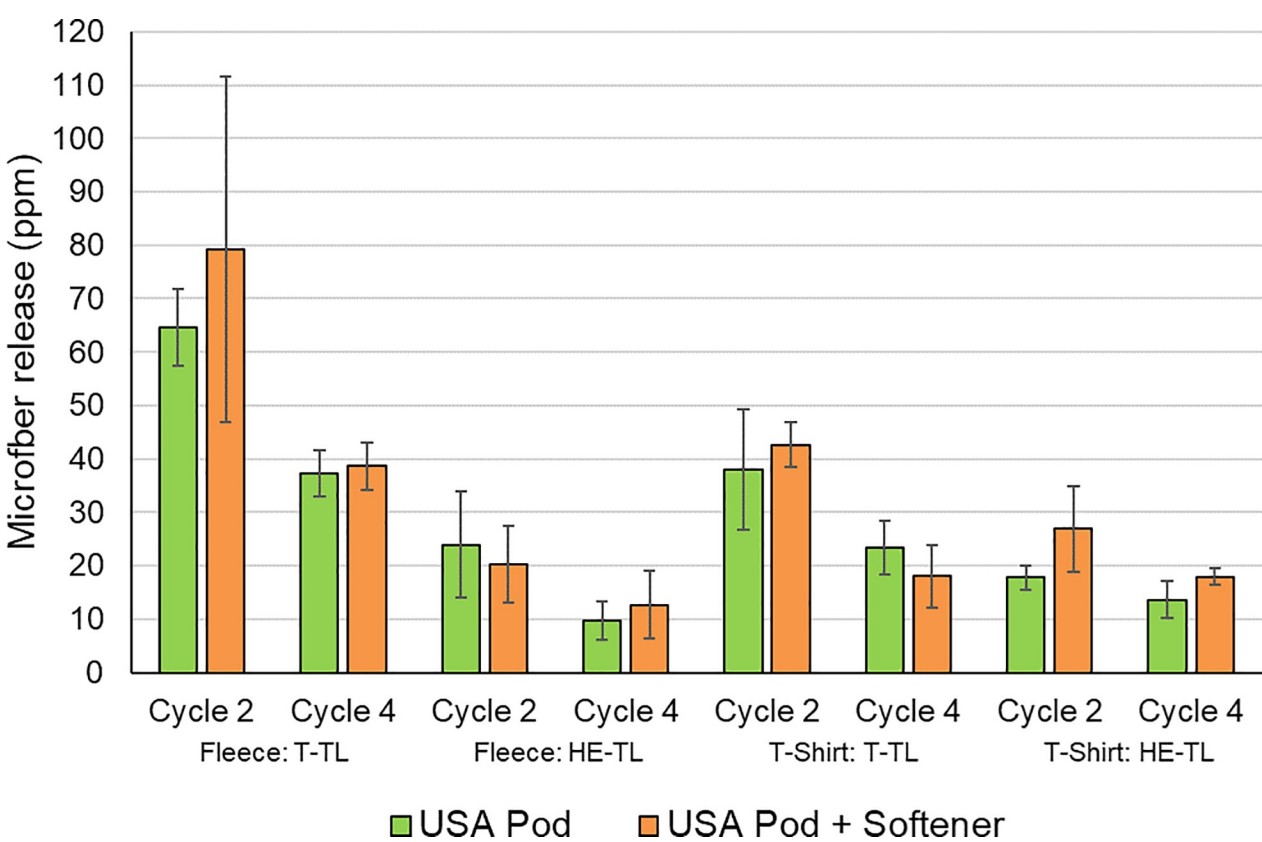

**Fig 8. Impact of fabric softener on microfiber release from polyester fleece and performance T-shirt (n = 64).** U.S.A. Traditional Top Loader (T-TL) and High-Efficiency Top-Loader (HE-TL) conditions. Error bars represent the standard deviation.

be more easily released compared to synthetic fibers which are often present in the form of filament yarns. Microfiber release can be significantly reduced using colder and quicker washing cycles and washing complete but not over-filled loads. Such interventions have other garment-care benefits that will enhance the useful lifetime of clothing with broader environmental impact benefits. Reduced whiteness loss was measured in the present study although colder and quicker washing cycles are also likely to have other benefits including reduced dye fading, reduced pilling and abrasive damage, and reduced incidence of holes and tears. Reducing such fabric integrity loss could lead to fewer garments being discarded [35] and longer garment lifetimes, leading to significantly reduced environmental impact.

The main barrier to consumers adopting colder and quicker washing cycles is concern that they will not achieve the desired level of cleaning and malodor removal, calling for further advances in cold water cleaning technology such as enzymes to eliminate the performance gap between conventional warm cycles and colder and quicker alternatives.

In North America, where the appliance market is currently divided between traditional top-loading washing machines and emerging High-Efficiency machines with lower water volumes, results are supportive of converting to High-Efficiency appliances as a way of significantly reducing microfiber generation in addition to the other environmental benefits of High-Efficiency washing machines such as 35–50% less water consumption and about 50% less energy per load [36].

The results from testing under European and North American conditions suggest that high water to fabric ratio is an important driver of microfiber loss, likely driven by increased flow of

water through yarns and hence increased fiber extraction. This provides an additional reason why the appliance industry should continue to develop approaches to reduce water consumption in laundry.

Our results also found that the use of detergent pods or fabric softener had no impact on microfiber release, and that garments showed the highest level of release in the first few washes and did not increase after a long multicycle wash history. Although fabric softener did not have any direct impact on microfiber loss, further work is needed to fully understand whether use of this product could lead to a significant indirect reduction in microfiber levels by extending the lifetime of textiles and reducing their level of replacement with new items that will cause a high level of microfiber release in their first few wash cycles. Additional work is also needed to fully understand the impact of powdered laundry detergents on microfiber release. From a methodology standpoint this is more challenging as these products are inherently less soluble than liquid detergents which makes gravimetric analysis of released fibers more difficult. Although there are several compositional differences between powder detergents and liquids, including the presence of alkalinity sources and bleach in the former, these are unlikely to exacerbate the fundamental fiber unplucking and breakage mechanisms involved in microfiber release, but this needs to be confirmed.

The apparent contradiction of the present results with some earlier reports is likely to be driven by differences in test methodology, particularly in that the present study was completed using only sets of full-scale washing machine rather than miniaturized model systems such as the GyroWash, Washtec or Launder-Ometer® and did not involve any artificial garment aging processes. This suggests that further work is needed to develop and validate appropriate test methods against the reality of consumer washing conditions and garment aging, especially if they will form the basis of further improvements in textile design, appliance design, or recommendations to consumers about how they can reduce their impact on microfiber pollution through choices relating to laundry products and processes, including choice of appliances and their programs. Different methods may be justified for textile comparisons (e.g. involving tests without detergents) and studies involving different fabric care products and washing conditions. Therefore, it is important that any new methods emerging in this area come with a clear scope of application and are appropriately validated on this basis.

Finding an ultimate solution to the pollution of marine ecosystems by microfibers released during laundering will likely require significant interventions in both textile manufacturing processes (e.g. to reduce the inherent release profile of items, and/or conduct filtered pre-washing to remove the most labile fibers) and washing machine appliance design (e.g. to introduce filtering systems). In the meantime, consumers can be assured that their continued use of liquid detergents and fabric softeners will not exacerbate the issue, and that they can significantly reduce their contribution to the issue by washing in colder and quicker cycles, washing complete but not overfilled loads and, in North America, transitioning to High-Efficiency washing machines.

## Supporting information

**S1 Fig. Images of typical real wash loads.**
(DOCX)

**S2 Fig. Light microscopy images of example filtered fibers.**
(DOCX)

**S3 Fig. Boxplot of microfiber release from 79 European soiled consumer wash loads (n = 79).**
(DOCX)

**S1 Table. List of whiteness monitor fabrics.**
(DOCX)

**S2 Table. Load masses and microfiber masses for 79 European soiled consumer wash loads (n = 79).**
(DOCX)

**S3 Table. Digestion of fibers.**
(DOCX)

**S4 Table. Microfiber release from soiled consumer wash loads in a 40˚C cycle (n = 19) with cold express cycle (n = 19).**
(DOCX)

**S5 Table. Comparison of whiteness loss (ΔE2000) from 40˚C (n = 19) with cold express cycle (n = 19) on 18 fabric types.**
(DOCX)

**S6 Table. Impact of using European detergent pod on microfiber release from polyester fleece (n = 24).**
(DOCX)

**S7 Table. Impact of fabric softener on microfiber release from polyester fleece: European conditions (n = 36).**
(DOCX)

**S8 Table. Impact of fabric softener on microfiber release from polyester fleece and performance T-shirt (n = 64). U.S.A.** Traditional Top Loader (T-TL) and High-Efficiency Top-Loader (HE-TL) conditions.
(DOCX)

## Acknowledgments

The authors sincerely thank Liam Medina, Blessing Adetayo Dina, Yvonne McMeekin, Hannah Whiteside and Claire Morley for assistance in the collection of experimental data.

## Author Contributions

**Conceptualization:** Neil J. Lant.

**Data curation:** Neil J. Lant, Adam S. Hayward, John R. Dean.

**Formal analysis:** Neil J. Lant.

**Funding acquisition:** Neil J. Lant.

**Investigation:** Adam S. Hayward, Madusha M. D. Peththawadu, Kelly J. Sheridan.

**Project administration:** Neil J. Lant, Adam S. Hayward, John R. Dean.

**Supervision:** Neil J. Lant, Kelly J. Sheridan, John R. Dean.

**Writing – original draft:** Neil J. Lant, Adam S. Hayward, Madusha M. D. Peththawadu, Kelly J. Sheridan, John R. Dean.

**Writing – review & editing:** Neil J. Lant, John R. Dean.

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
