## [Decision Letter · Decision Letter 0]

3 Feb 2020

PONE-D-19-32053

Microfiber Release from Real Soiled Consumer Laundry and the Impact of Fabric Care Products and Washing Conditions

PLOS ONE

Dear Dr Lant,

Thank you for submitting your manuscript to PLOS ONE. After careful consideration, we feel that it has merit but does not fully meet PLOS ONE’s publication criteria as it currently stands. Therefore, we invite you to submit a revised version of the manuscript that addresses the points raised during the review process.

We would appreciate receiving your revised manuscript by Mar 19 2020 11:59PM. To enhance the reproducibility of your results, we recommend that if applicable you deposit your laboratory protocols in protocols.io, where a protocol can be assigned its own identifier (DOI) such that it can be cited independently in the future. For instructions see: http://journals.plos.org/plosone/s/submission-guidelines#loc-laboratory-protocols

We look forward to receiving your revised manuscript.

Kind regards,

Pratheep K. Annamalai

Academic Editor

PLOS ONE

Journal Requirements:

2. We note that Figures 1 and S1 in your submission contain copyrighted images. All PLOS content is published under the Creative Commons Attribution License (CC BY 4.0), which means that the manuscript, images, and Supporting Information files will be freely available online, and any third party is permitted to access, download, copy, distribute, and use these materials in any way, even commercially, with proper attribution. For more information, see our copyright guidelines: http://journals.plos.org/plosone/s/licenses-and-copyright.

1.         You may seek permission from the original copyright holder of Figures 1 and S1 to publish the content specifically under the CC BY 4.0 license.

3. Please clarify whether there was any ethical oversight over the study, and whether participants gave consent to take part in the study.

4. Thank you for stating the following in the Financial Disclosure section:

"This work was sponsored by Procter & Gamble, a manufacturer of detergents and fabric softeners. The funders provided support in the form of salaries for NJL and ASH."

We note that one or more of the authors have an affiliation to the commercial funders of this research study : Procter & Gamble.

Reviewers' comments:

Reviewer's Responses to Questions

**Comments to the Author**

1. Is the manuscript technically sound, and do the data support the conclusions?

Reviewer #1: Yes

Reviewer #2: Yes

2. Has the statistical analysis been performed appropriately and rigorously? 

Reviewer #1: I Don't Know

Reviewer #2: No

3. Have the authors made all data underlying the findings in their manuscript fully available?

Reviewer #1: Yes

Reviewer #2: No

4. Is the manuscript presented in an intelligible fashion and written in standard English?

Reviewer #1: Yes

Reviewer #2: Yes

5. Review Comments to the Author

Reviewer #1: The manuscript described the microfiber release from textiles. The topic is current and is having an increasing number of studies in the last few years. Authors should more clearly differentiate their study from those published before.

1. More information about textile characteristics of yarn, fabric constructions, pilling, etc., should be added;

2. The power of wahsing machine is very important. Please add it;

3. Please add the washing procedure each time;

4. The fiber characteristics shoud be added in the result section;

5. Refine the conclusion section.

Reviewer #2: Abstract

Line 26 - "an average of over 100 ppm" - tell us what the average was, not a number that was below the average, also include the uncertainty and n = X

Line 28 - 30 - give actual release data including uncertainties and either the p value that indicates a significant difference or don't mention the difference if it isn't significant

Line 30 - 32 - what are quicker and colder washes - better to be quantitative

Line34 - 36 - what about powder detergents which, I would guess, are probably more regularly used than pods or plain water

LIne 41 - 44 - does colder quicker washes reduce the total fibre release or the fibre release per wash, what was the low level of fibre release?

Methods

Line 155 - any particular reason why you didn't use washing powder but used a liquid detergent - is this based on industry data on frequency of use of these different detergents?

Line 173-4 - why weren't the fabrics dried between washes - surely that would reflect normal use and it could be that the wetting / drying cycle has a significant impact on fibre production.

Line 186-188 - doesn't whether microfibres are collected by the drier or not depend on the type of tumble drier used? Don't condensing tumble driers also collect water which microfibres could end up in via suspensions?

Line 211 - none the less in terms of quality control shouldn't you confirm in your study that this wash process is also working as presumably different machines / fabrics may behave differently?

Lines 248 - 251 - what shapes and characteristics features related to synthetic fibres and which to cotton and wool?

Line 253 - how was refractive index mesaured?

From line 267 - this is about recovery of fibres from filter paper but haven't some fibres already been removed using tape as detailed in the previous section?

Line 27 4 - what was the purpose of the sodium hypochlorite?

Fig 2. R2 value of curve needs a p value to be meaningful - also need to state the type of equation used to fit the trend line

Line 316 - 318 - give uncertainties and do stats. to see if these are significantly different.

Line 330 - 333 - you need to make it clear here that you are assuming that your digestion dissolved e.g. the cotton and viscose fibres rather than testing for this and that you are relying on the reference [31] for this. I think these statements about what the digestion procedure digests belong in your methods since they are not your results. Also it isn't clear how you generated the data for Fig. 3 - did you weigh the solid residue between digestions?

Ine 341 - won't this depend on the fabric composition?

LIne 353 - unwise to rely on anecdotal evidence in a scientific publication. I would leave it that you don't know and didn't test the colourless fibres

Line 358 - again, relying on anecdotal evidence - be clearer what this is here if you retain this.

LIne 376 here you mention statistics when comparing two sets of data. Does that mean where you don't mention them the differences weren't significant. If you mention differences which weren't statistically significant elsewhere really you should remove these. The whole point of the stats. is that they allow you to know whether a difference is meanfingful or not, so if a difference isn't significant you can't say there is a difference.

Fig. 4 - worth stating what the various lines on th ebox plot mean as different people use different protocols, e.g. means vs medians

LIne 405 - why in the absence of detergent - is that typical behaviour?

Fig. captions - as well as stating what the error bars are you need to say what n =

Data is given in graphs but needs to be provided in tabular form without averaging in the Supplementary information for ease of use by others.

6. PLOS authors have the option to publish the peer review history of their article (what does this mean?). If published, this will include your full peer review and any attached files.

Reviewer #1: Yes: Lihui An

Reviewer #2: No

---

## [Author Response · Author response to Decision Letter 0]

26 Mar 2020

Dear Dr P. K. Annamalai,

Re: Microfiber Release from Real Soiled Consumer Laundry and the Impact of Fabric Care Products and Washing Conditions – Response to Reviewers

 We thank the academic editor and reviewers for their careful consideration of our article and the feedback provided. A comprehensively revised version of the MS now been submitted, and as requested, we have provided clean and marked-up versions of the revised version. Each point raised by the reviewers has either been reflected in the amended MS and/or discussed in this letter. Taking each point from the decision letter in turn:

• We have ensured that our manuscript meets PLOS ONE's style requirements, including those for file naming.

• Figures 1 and S1 have been revised to ensure they are free of any copyrighted images. Specifically, the equipment images in Figure 1 have been replaced by those taken by the authors. All images in S1 were recorded by the authors and those images which included brand names have been removed. We certify that all images are free to be published under the Creative Commons Attribution License (CC BY 4.0). All figure files were confirmed to meet PLOS’s requirements using the PACE digital diagnostic tool.

• Regarding ethical oversight, no humans or animals were involved in the study beyond those scientists directly involved in the work. Laundry loads were borrowed from volunteers on the understanding that they would be used in scientific research and returned to the volunteers after washing.

• We have amended the funding statement using the suggested text, as follows: 

“The funder provided support in the form of salaries for N.J.L. and A.S.H. and decision to publish but did not have any additional role in the study design, data collection and analysis or preparation of the manuscript. The specific roles of these authors are articulated in the ‘author contributions’ section. This does not alter our adherence to PLOS ONE policies on sharing data and materials.”

• The entire data set has now been included as supplementary tables. Thus, every data point used to generate each graph has now been presented in full. 

The following comments relate to the reviewers’ responses to questions:

Q1: No response needed as both reviewers agreed that the manuscript is technically sound and that the data supports the conclusions. 

Q2: No response needed as no issues regarding statistical analysis were raised

Q3: One reviewer commented that all data underlying the findings in the manuscript was not available. All data has now been included in the form of supplementary tables. 

Q4: No response needed as no issues regarding presentation and language were raised. The MS has been proof-read to eliminate typographical and grammatical errors. 

Q5: The miscellaneous comments in this section have been addressed in the table below: 

Comment Response

The manuscript described the microfiber release from textiles. The topic is current and is having an increasing number of studies in the last few years. Authors should more clearly differentiate their study from those published before.

1. More information about textile characteristics of yarn, fabric constructions, pilling, etc., should be added;

2. The power of washing machine is very important. Please add it;

3. Please add the washing procedure each time;

4. The fiber characteristics should be added in the result Refine the conclusion section. 1. More textile characteristics from the supplier specification have been included. 

2. In European and North America (focus of these studies), power is not a distinguishing feature of any washing machines. It may be more important in markets with semi-automatic appliances. Power consumption data of European and North American washing machines is published in the manuals of these appliances, but this isn’t relevant. As full details of appliance models are provided, anyone requiring further information about any aspect of the washing machine can consult the manuals online. 

3. This has been done. It was missing in one instance and in all cases reference is made to the relevant general method in the materials and methods section.

4. Done. Conclusion has also been sharpened up.

Line 26 - "an average of over 100 ppm" - tell us what the average was, not a number that was below the average, also include the uncertainty and n = X Done

Line 28 - 30 - give actual release data including uncertainties and either the p value that indicates a significant difference or don't mention the difference if it isn't significant Done

Line 30 - 32 - what are quicker and colder washes - better to be quantitative Done

Line 34 - 36 - what about powder detergents which, I would guess, are probably more regularly used than pods or plain water In Europe and North America, the focus of the present study, liquid detergents are much more popular than powders which are rapidly declining in market share. We may include powders in a future article focused on other markets such as Asia, Africa and Latin America where powder detergents are still the most popular detergent form.

Line 41 - 44 - does colder quicker washes reduce the total fibre release or the fibre release per wash, what was the low level of fibre release? Updated to be more specific.

Line 155 - any particular reason why you didn't use washing powder but used a liquid detergent - is this based on industry data on frequency of use of these different detergents? Yes. See above. Powders were recently described as ‘almost extinct’ in the USA and are declining in other regions. See this link for further details.

https://cen.acs.org/business/consumer-products/Almost-extinct-US-powdered-laundry/97/i4

Line 173-4 - why weren't the fabrics dried between washes - surely that would reflect normal use and it could be that the wetting / drying cycle has a significant impact on fibre production. Line drying is the most common method of drying laundry outside North America. There is no literature precedent for indoor line drying causing significant fiber loss, so testing without such a line drying step is expected to yield similar results to continuous cycling in the absence of such a drying process. The same is not true for tumble drying as the air flow causes significant fiber loss, which is why the North America – focused testing included tumble drying.

Line 186-188 - doesn't whether microfibres are collected by the drier or not depend on the type of tumble drier used? Don't condensing tumble driers also collect water which microfibres could end up in via suspensions? All tumble dryers will collect fibers in the lint filter, and these are disposed in dry household refuse so have not been implicated as a source of marine pollution. We have not encountered fibers in condenser dryers, but in any case, that type of appliance was not used in the present study. We chose typical tumble-drying conditions for the present study; a study of impact of tumble-drying conditions on subsequent microfiber loss in the wash process was not included.

Line 211 - none the less in terms of quality control shouldn't you confirm in your study that this wash process is also working as presumably different machines / fabrics may behave differently? This has now been clarified. Rationale for not repeating the detail is that Kelly et al used the same washing machines and some of the same textiles as the present study.

Lines 248 - 251 - what shapes and characteristics features related to synthetic fibres and which to cotton and wool? This has now been explained.

Line 253 - how was refractive index measured? This section has been rewritten and a reference added.

From line 267 - this is about recovery of fibres from filter paper but haven't some fibres already been removed using tape as detailed in the previous section? The protocol has been clarified with addition of a new paragraph.

Line 274 - what was the purpose of the sodium hypochlorite? This has now been explained.

Fig 2. R2 value of curve needs a p value to be meaningful - also need to state the type of equation used to fit the trend line Done. This has now been fully explained with requested data and equation in the text. It is now unambiguously clear that a squares equation has been applied.

Line 316 - 318 - give uncertainties and do stats. to see if these are significantly different. This has been fully updated with stats.

Line 330 - 333 - you need to make it clear here that you are assuming that your digestion dissolved e.g. the cotton and viscose fibres rather than testing for this and that you are relying on the reference [31] for this. I think these statements about what the digestion procedure digests belong in your methods since they are not your results. Also it isn't clear how you generated the data for Fig. 3 - did you weigh the solid residue between digestions? This has indeed been moved to the methods section and clarified.

Line 341 - Won't this depend on the fabric composition? Absolutely. That is why we tested real mixed loads from consumers – that is an important novel aspect of our study. On the face of it, the high proportion of natural fibers in the released fiber mass is inconsistent with the high incidence of synthetic fibers in the general clothing market. This can be rationalized by the fact that most natural yarns are produced by spinning relatively short staple fibers whereas many synthetic yarns are spun from long filaments that are inherently more resistant to release. This point is discussed in the conclusions.

Line 353 - Unwise to rely on anecdotal evidence in a scientific publication. I would leave it that you don't know and didn't test the colourless fibres. Agree. That section has been updated accordingly.

Line 358 - Again, relying on anecdotal evidence - be clearer what this is here if you retain this. This sentence has been removed.

Line 376 Here you mention statistics when comparing two sets of data. Does that mean where you don't mention them the differences weren't significant. If you mention differences which weren't statistically significant elsewhere really you should remove these. The whole point of the stats. is that they allow you to know whether a difference is meaningful or not, so if a difference isn't significant you can't say there is a difference. We describe significant differences, with supporting stats, in relation to the comparisons in Figures 4 and 5. We state that there are no significant differences between the comparisons involved in Figures 6 and 7, and this is evident from the standard deviations show in the graphs and additional data that is now included as supplementary tables. 

Fig. 4 - Worth stating what the various lines on the box plot mean as different people use different protocols, e.g. means vs medians. The box plot has now been fully described at first mention, which is the reference to S5 Fig.

Line 405 - Why in the absence of detergent - is that typical behaviour? It is not typical for consumers to wash in the absence of detergent. The only purpose of making this comparison is to understand, on a single-variable basis, whether the detergent influences microfiber release. 

Fig. captions - as well as stating what the error bars are you need to say what n = Done

Data is given in graphs but needs to be provided in tabular form without averaging in the Supplementary information for ease of use by others. Done

We believe that the revisions have significantly improved the MS and hope that they will be sufficient to enable acceptance of the article. We look forward to hearing from you.

Yours sincerely,

Dr Neil J. Lant, on behalf of the authors.

---

## [Editor Report · Decision Letter 1]

5 May 2020

Microfiber Release from Real Soiled Consumer Laundry and the Impact of Fabric Care Products and Washing Conditions

PONE-D-19-32053R1

Dear Dr. Neil Lant

Thank you again for submission to PLOS One and taking time for carefully revising the manuscript according the comments from reviewers. I am pleased to inform you that your revised manuscript has been judged scientifically suitable for publication and will be formally accepted for publication once it complies with all outstanding technical requirements.

With kind regards,

Pratheep K. Annamalai

Academic Editor

PLOS ONE

---

## [Editor Report · Acceptance letter]

11 May 2020

PONE-D-19-32053R1 

Microfiber Release from Real Soiled Consumer Laundry and the Impact of Fabric Care Products and Washing Conditions 

Dear Dr. Lant:

I am pleased to inform you that your manuscript has been deemed suitable for publication in PLOS ONE. Congratulations! Your manuscript is now with our production department. 

With kind regards,

on behalf of

Dr. Pratheep K. Annamalai 

Academic Editor

PLOS ONE